# Lightning Cessation Guidance Using Polarimetric Radar Data and Lightning Mapping Array in the Washington, D.C. Area

John J. Drugan and Ari D. Preston *

Department of Atmospheric Sciences, Northern Vermont University-Lyndon, Lyndonville, VT 05851, USA; jdrugan@versar.com
*   Correspondence: aaron.preston@northernvermont.edu

**Abstract:** Polarimetric radar data and total lightning data are used to develop lightning cessation guidance for isolated cells in the Washington, D.C. area. A total of 23 non-severe thunderstorms during the 2015–2017 warm seasons are analyzed. Radar and lightning data are superimposed using the Warning Decision Support System–Integrated Information software to develop cessation algorithms. This includes using the hydrometeor classification algorithm to locate graupel for each convective cell. Results show that the three best-performing cessation algorithms use thresholds of (1) $Z_H \geq 40$ dBZ at $-5$ °C, (2) $Z_H \geq 35$ dBZ at $-10$ °C, and (3) graupel at $-15$ °C. Lightning is not expected 15 min after the threshold is no longer met for each algorithm. These algorithms are recommended only for isolated cells in the Washington, D.C. area. Further study needs to be completed to draw conclusions for other convective cell types and different geographic regions.

**Keywords:** lightning; radar; forecasting





## 1. Introduction

Lightning ranks as one of the deadliest weather phenomena. Each year, 50 casualties and hundreds of injuries result from lightning strikes in the United States [1]. The importance of forecasting lightning for the safety of the public is crucial, as well as cost effective for outdoor businesses [2]. Most deaths occur before or mainly after peak lightning activity [3]. This is because these flashes occur at times when lightning has not yet become threatening (e.g., first flash) or is no longer deemed threatening (e.g., last flash) [2]. The most commonly used safety rule for lightning is the 30–30 rule [3], where the second "30" refers to the wait time in minutes before resuming outdoor activities after the most recent lightning flash. However, 30 min is a lengthy amount of time to wait and can delay business operations including aviation, construction, sports, and community events [4]. Furthermore, additional flashes can sometimes be observed even after waiting longer than 30 min of no lightning activity [5]. Therefore, developing lightning cessation guidance can assist outdoor safety measures.

The noninductive charging process for isolated cells involves the electrification of individual hydrometeors within the cloud and is independent of nearby electrical fields [6]. Graupel, ice crystals, and supercooled water are the primary hydrometeors involved in this process where gravity and updrafts help to produce charge separation [7,8]. When ice crystals collide with graupel in the presence of supercooled water, the graupel will obtain a negative charge while the ice crystals obtain a positive charge [9]. Graupel primarily obtains this charge in the mixed phase region of the cloud between 0 °C and $-20$ °C. Positive charges are observed in temperatures warmer than 0 °C [7,10]. Updrafts carry the lighter ice crystals to greater heights than the heavier graupel. This results in a tripole charge structure with a positive layer at the top of the cloud, a negative layer of graupel in the middle, and another weaker positive charge at the bottom of the cloud with warmer graupel [11]. This charge separation eventually results in the first flash of a storm being produced, as well as subsequent flashes [12]. A study by [5] showed that this mixed phase

region is a crucial component of developing successful algorithms for lightning cessation guidance.

Several studies have already focused on lightning cessation guidance using different variables and thresholds [5,13–19]. Some of these studies were more effective than others, but non-isolated cells proved ineffective for cessation guidance due to the charging of nearby cells [5,13]. Studies by [14,15] examined radar differential phase signatures and vertical alignment of ice crystals near the end of lightning activity. They found that certain alignments, shapes, sizes, and densities of ice crystals are present for cloud electrification among larger horizontal reflectivity values. Furthermore, [16] correlated the timing of flashes with vertically integrated ice crystals as guidance for lightning cessation. A study by [17] used maximum interval guidance where a set time limit between flashes would determine the possibility of another flash. The study found that most storms did not produce another flash after 10 min of no lightning activity. However, [5] found cases in which flashes were produced after 10 min of no lightning activity using polarimetric radar data in the Cape Canaveral area of Florida. Their study examined the presence of graupel at specific temperature levels in non-severe, isolated storms. The criterion for isolation was defined as a cell being separated by less than 15 dBZ composite reflectivity (i.e., $Z_H < 15$ dBZ) from any other cells present. They determined that lightning advisories could be lifted after a 10 min absence of (1) graupel at $-10\,^{\circ}$C and (2) $Z_H \geq 35$ dBZ at $-10\,^{\circ}$C. Their cessation algorithm was only successful for isolated cells since non-isolated cells could be charged by nearby storms and experience additional flashes even after graupel was no longer present in the parent storm. A study by [13] attempted to create cessation algorithms for non-isolated storms but found that their algorithms were ineffective. More recently, [18] examined convective cells in the Washington, D.C. area to develop a bootstrap model for lighting cessation guidance after its proven success for storms in Florida [19]. However, the change in geographic region made their cessation algorithm less effective than in the tropics. They found that thresholds designed to operate in Florida ended up being lower in Washington, D.C. The findings in [18] demonstrate that previous cessation guidance needs to be adjusted when examining a different geographic climate. The presence of graupel diminished from upper levels to lower levels as lightning ended, and their model predicted lightning cessation later than when it actually occurred. They also determined that the existence of graupel at specified isotherm levels in the mixed phase region played a crucial role in cloud electrification.

The current study builds on that of [5] by developing wait time algorithms for lightning cessation guidance, but in a different geographic region. This will help us determine how effective a wait time algorithm approach is for cessation guidance in an environment outside of central Florida. The goal is to develop comprehensive cessation guidance with a tool that is easily adaptable to other geographic areas. This study will provide insight into the feasibility of this and future operational utility. In fact, [5] recommended testing the effectiveness of cessation algorithms in an area like Washington, D.C., due to its 3D lightning network. The current study examines 23 isolated cells in the Washington, D.C. area. Similar to [18], a different geographic region may require an algorithm adjustment for effective cessation guidance as regional variations in thermodynamic environments can alter the charge structure in a storm [20]. Freezing levels are found at varying altitudes depending on tropospheric temperatures [21]. Therefore, the presence of graupel at $-10\,^{\circ}$C may be less effective for cessation algorithms in the Mid-Atlantic region compared to Florida. For example, Washington, D.C., (23.3 $^{\circ}$C) has a 5-month warm season (May-September) average surface temperature that is 3.5 $^{\circ}$C colder than the average surface temperature in the Cape Canaveral area of Florida (26.8 $^{\circ}$C) during the warm season. This suggests that Washington, D.C., will have lower freezing levels that could change the effectiveness of algorithms for cessation guidance. Furthermore, different modes of convection also could impact cessation algorithms. For example, sea breeze fronts are the dominant mechanism triggering non-severe cells in Florida during the warm season [22]. Meanwhile, thunderstorms in Washington, D.C., often develop along northwesterly synoptic boundaries [23]. The

Washington, D.C. area also has higher aerosol concentrations than Florida (due to more pollution), which could impact charging mechanisms in electrified storms as suggested by [18].

## 2. Data and Methods

### 2.1. Lightning Data

The Washington, D.C., Lightning Mapping Array (DCLMA) was developed by the National Aeronautics and Space Administration, the National Oceanic and Atmospheric Administration, and New Mexico Institute of Mining and Technology [24]. It is made up of 10 lightning detectors covering ~7000 km$^2$ that detect very high frequency (VHF) radiation emissions produced when in-cloud (IC) or cloud-to-ground (CG) lightning changes direction or speed and maps the data on a 3D-plane using a Global Positioning System [24,25]. The total lightning (IC and CG) data gathered allows for intervals of 1 min to be displayed on a user interface. Days with lightning in the Washington, D.C. area were determined by examining archives of DCLMA data.

The Warning Decision Support System–Integrated Information (WDSS−II) [26] was used to overlay radar data from Sterling, VA, (KLWX) and Dover, DE, (KDOX) and lightning data from the DCLMA at different temperature levels. This software also utilizes hydrometeor classification to identify the presence of graupel at temperature levels crucial to storm electrification. This information was used to develop an appropriate threshold for lightning cessation guidance.

### 2.2. Radar and Environmental Data

Archived radar images from the University Corporation for Atmospheric Research were used to determine dates where isolated cells (white circles; Figure 1) were present on days with lightning. Using similar criteria as [5], we defined isolation as a cell being separated by composite $Z_H < 15$ dBZ from nearby cells. All convective cells in this study had to remain isolated through lightning cessation and storm dissipation. A total of 23 isolated storms were selected during the 2015–2017 warm seasons.

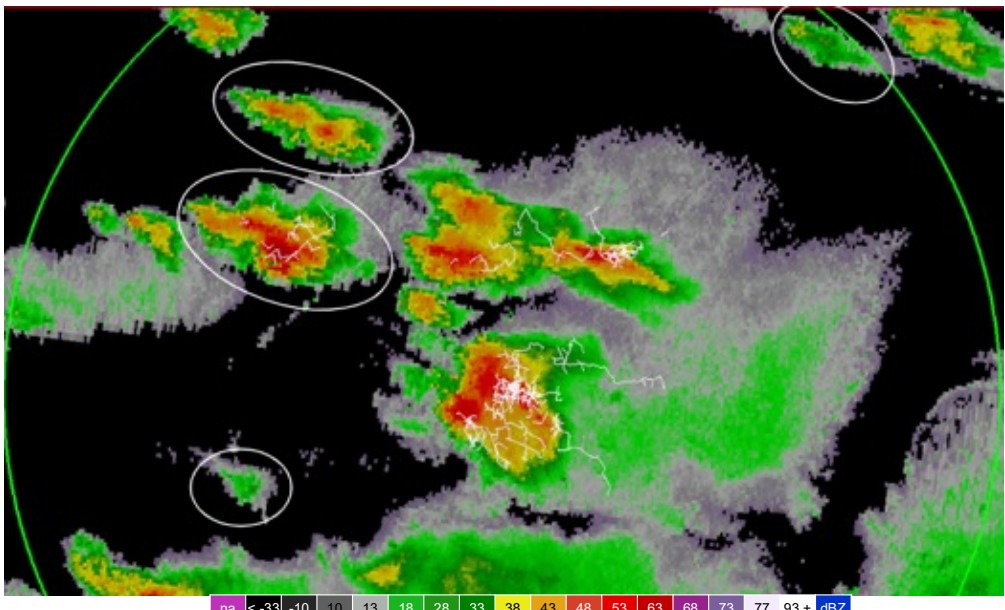

**Figure 1.** Composite reflectivity with 3D lightning channels overlaid in WDSS−II at 2153 UTC 21 June 2016. Examples of isolated cells are highlighted with white circles.

Polarimetric radar data were collected from the WSR-88D S-band radars KLWX and KDOX. Using a conversion algorithm (ldm2netcdf) [26], radar data were converted to NetCDF format so they could be used within the WDSS−II software [26]. A radius of

150 km was created around the DCLMA (Figure 2) as the area of focus since lightning detection efficiency within this region is greater than 90% [27]. All storms also occurred within 200 km of KLWX or KDOX radar sites as shown in Figure 2.

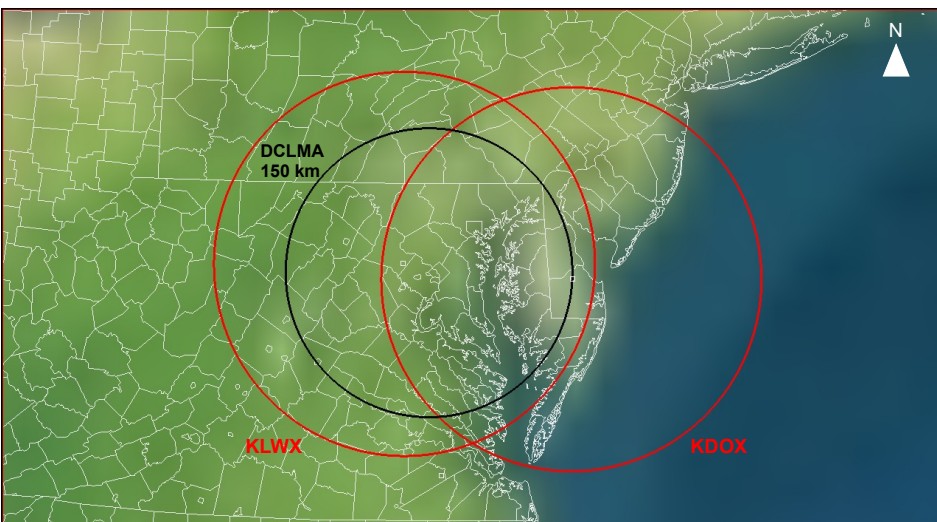

**Figure 2.** The red circles extend 200 km from the KLWX and KDOX radar sites. All 23 non-severe storms were within 150 km (black circle) of the DCLMA center.

The Rapid Refresh model (RAP) [28] data were used in this study and had to be converted to the proper format using the NetCDF algorithm [26]. To finalize processing, the near storm environment algorithm (nse) [26] was run to obtain RAP temperature level altitudes (i.e., 0 °C, −5 °C, −10 °C, −15 °C, and −20 °C) at the time of each convective cell. Running the merging algorithm (w2merger) [26] then allowed WDSS−II to blend radar data (~1 km horizontal and vertical grid spacing) between KLWX and KDOX using an exponential weighting function based on distance to each radar. The merging algorithm also decreased the time interval from 5 min to 1 min as each elevation scan is processed. The WDSS−II software does not wait for full radar scans to be completed. The merging algorithm was run alongside the simulator algorithm (w2simulator) [26] to allow chrono- logical files to be processed and viewed within WDSS−II. We also ran the hydrometeor classification algorithm (HCA; w2dualpol) [26], which is based on the algorithm developed by J. Krause at the National Severe Storms Laboratory and described by [29–31]. The algo- rithm uses polarimetric base products, reflectivity information, as well as texture products along the radar beam. The HCA used here identifies hydrometeor types and assigns a particle identification value to the dominant or majority quantity within each radar range gate of a volume scan [32]. As with the other radar parameters, WDSS−II interpolates the HCA categories to ~1 km horizontal and vertical grid spacing. The HCA category at a location is the particle identification value from the closest radar (not a weighting function based on distance to each radar). This product was used to determine the presence of graupel in the mixed phase region of a storm (Figure 3).

### 2.3. Lightning Flash Algorithm

Raw lightning data were retrieved from the DCLMA at 1 min intervals using the lightning ingest algorithm. This algorithm gathers data and creates an index for process- ing (w2lmaflash) [26]. Lightning flash density products were created using the lightning algorithm (w2ltg) [26] and these data were assembled to view 3D lightning channels and flash initiation points for isolated cells. We used the default flash algorithm thresholds, where each VHF source had to be within 300 ms and 5 km from one another [26]. Fur- thermore, flashes also had to have a minimum of three VHF sources to eliminate potential noise [33,34]. The lightning products were then added to the WDSS−II software and overlaid on radar data at 1 min intervals (Figure 4).

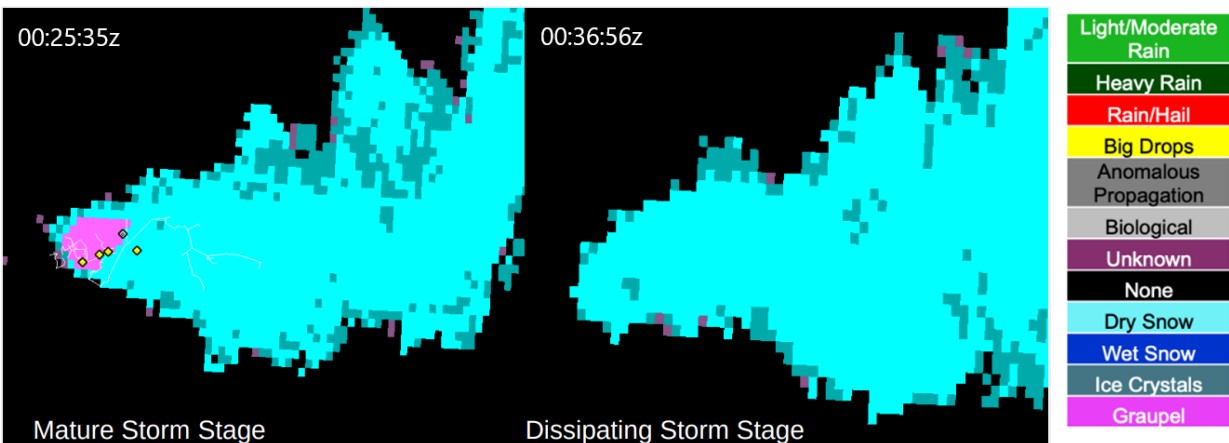

**Figure 3.** The HCA displays the most likely hydrometeor type based on a combination of polarimetric radar products. These two panels show the dissipation of graupel at −10 °C in a non-severe, isolated storm on 22 June 2016. The yellow diamonds represent the initiation location of flashes detected by the DCLMA within 1 min of the radar time stamp.

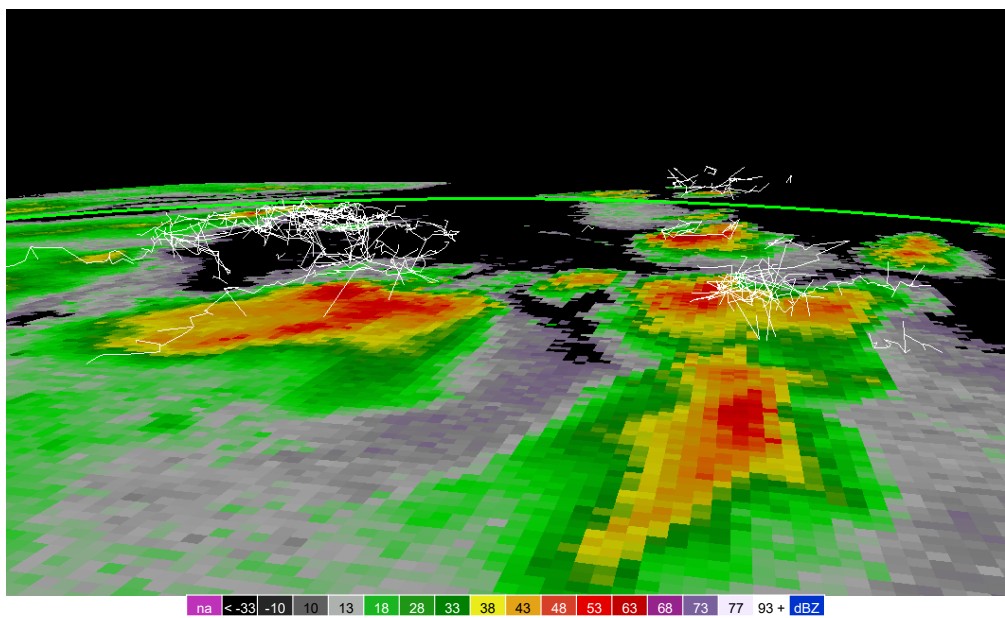

**Figure 4.** Side view of 3D lightning channels overlaid on composite reflectivity in WDSS−II at 2153 UTC 21 June 2016.

### 2.4. Simulating Case Studies

Each case study had to remain isolated through the end of its lifetime. A total of 23 non-severe, isolated cells were analyzed and lightning advisories were simulated for each event. We defined a non-severe storm as one without a recorded severe weather report [35]. The simulated lightning advisory begins when the first lightning flash is detected. The advisory ends after waiting a certain amount of time without meeting a radar-based threshold. The five thresholds used were:

1.  $Z_H \geq 40$ dBZ
2.  $Z_H \geq 35$ dBZ
3.  presence of graupel
4.  presence of graupel with $Z_H \geq 40$ dBZ
5.  presence of graupel with $Z_H \geq 35$ dBZ

These thresholds focus on storm intensity ($Z_H$) and graupel presence, which are both critical to noninductive charging theory. The five thresholds above were tested at five different temperature levels (0 °C, −5 °C, −10 °C, −15 °C, and −20 °C) and three different wait times (5, 10, and 15 min) for a combination of 75 total algorithms. For each algorithm, once the threshold was no longer met, a wait time was started. If lightning cessation occurred before the wait time ended, it was considered a hit since the lightning advisory would have been lifted after lightning activity ended. If lightning activity was ongoing when the lightning advisory ended, it was considered a false alarm as the advisory would have been lifted before lightning cessation. False alarms are especially dangerous since they convey that it is safe to move outside before the last flash of the storm has occurred. If a radar-derived threshold was never met during the storm's lifetime, it was considered a missed event, meaning the advisory was never ended. We did not include null events since we only considered electrified storms, in which cessation must eventually occur. The 2 × 2 contingency table (Table 1) shows how each event was classified. Skill scores were calculated to determine the effectiveness of each algorithm as discussed below in Section 3.

**Table 1.** A 2 × 2 contingency table used to calculate skill scores for each cessation algorithm.

|  | Lightning Ended | Lightning Ongoing |
|---|---|---|
| **Advisory Ended** | Hit | False Alarm |
| **Advisory Ongoing** | Miss | Null |

## 3. Results and Conclusions

To select the best algorithm to use as a public lightning advisory, several statistics must be considered to eliminate ones that are unreliable, unsafe, have unreasonable advisory cancellation times, or lack time savings compared to the 30–30 rule. Performance metrics are based on applying the algorithms to the sample of 23 storms. Tables 2 and 3 show Probability of Detection (POD) and False Alarm Ratio (FAR), respectively. These skill scores were used to determine how well our cessation algorithms performed. Algorithms with POD < 1.0 had at least one missed advisory cancellation in our 23-storm dataset. A missed event means the lightning advisory never ended (i.e., it remained active through storm dissipation). We considered algorithms with missed events unreliable due to their inability to end advisories in a timely manner. Of the 75 algorithms tested, only 6 (8%) had a missed event (Table 2; POD < 1.0). Algorithms with false alarms (FAR > 0.0) are especially dangerous since they can be fatal. This is because a false alarm represents an algorithm that ends the advisory prematurely (i.e., before the last flash has occurred). This makes them unsafe for operational utility. Table 3 shows there were 28 (37%) algorithms with a false alarm (FAR > 0.0). These results demonstrate that cessation algorithms in Washington, D.C., struggle more with false alarms than missed events. This is consistent with [5] who also had more algorithms with a false alarm (9 out of 15; 60%) than a missed event (6 out of 15; 40%). Table 4 shows a performance metric that combines POD and FAR called Critical Success Index (CSI). Values in boldface represent CSI = 1.0, which indicates a very effective cessation algorithm (POD = 1.0, FAR = 0.0). The remaining algorithms in gray had CSI < 1.0 (28 out of 75; 37%), which means the algorithm had at least one false alarm and/or miss. These cessation algorithms are not considered further in this study and have been removed from Table 5.

Table 5 considers the timing of lightning advisory cancellations for the safest cessation algorithms from Tables 2–4. We seek cessation algorithms that balance safety and time savings. Table 5 shows the minimum, maximum, and average cancellation times for all 23 non-severe storms. Three algorithms stand above the rest (boldface; Table 5) based on the small variability of their advisory cancellation times. All three algorithms had a minimum cancellation time of 10 min after the last flash and maximum cancellation time of 45 min after the last flash. The rest of the algorithms shown (gray; Table 5) ended at least one lightning advisory less than 10 min after the last flash or greater than 45 min

after the last flash. If an advisory ends longer than 45 min after the final flash, it no longer provides time savings compared to the 30–30 rule. This resulted in 28 cessation algorithms being dismissed. Furthermore, it is important to note that if an advisory is ended less than 10 min from the previous lightning flash, the public may not feel safe to resume outdoor activities after having just observed a nearby flash. As a result, 16 additional cessation algorithms were disregarded due to this recent memory bias. One of the eliminated algorithms is the combined graupel and $Z_H \geq 35$ dBZ at $-10\,^\circ$C using a 10 min wait time as recommended in [5]. It should be noted that [5] did not eliminate algorithms based on a "too close for comfort" advisory cancellation time as done here. Regardless, this Florida-recommended cessation algorithm performed well in Washington, D.C., with a CSI = 1.0. However, the average advisory cancellation time for this cessation algorithm increased from 17.9 min (Florida) to 23.0 min (Washington, D.C.). The maximum advisory cancellation time increased even more from 31.0 min (Florida) to 44.0 min (Washington, D.C.). This trend of delayed advisory cancellations in Washington, D.C., (compared to Florida) is observed in many other cessation algorithms as discussed below.

**Table 2.** Probability of Detection defined by the total number of hits divided by the total number of hits and misses: POD = total hits/(total hits + total misses).

| Temp Level (°C) | Wait Time | dBZ $\geq$ 40 | dBZ $\geq$ 35 | Graupel | Graupel + dBZ $\geq$ 40 | Graupel + dBZ $\geq$ 35 |
|---|---|---|---|---|---|---|
| 0 | 5 min | 1.00 | 1.00 | 1.00 | 1.00 | 1.00 |
| | 10 min | 1.00 | 1.00 | 1.00 | 1.00 | 1.00 |
| | 15 min | 1.00 | 1.00 | 1.00 | 1.00 | 1.00 |
| −5 | 5 min | 1.00 | 1.00 | 1.00 | 1.00 | 1.00 |
| | 10 min | 1.00 | 1.00 | 1.00 | 1.00 | 1.00 |
| | 15 min | 1.00 | 1.00 | 1.00 | 1.00 | 1.00 |
| −10 | 5 min | 1.00 | 1.00 | 1.00 | 1.00 | 1.00 |
| | 10 min | 1.00 | 1.00 | 1.00 | 1.00 | 1.00 |
| | 15 min | 1.00 | 1.00 | 1.00 | 1.00 | 1.00 |
| −15 | 5 min | 0.94 | 1.00 | 1.00 | 1.00 | 1.00 |
| | 10 min | 0.95 | 1.00 | 1.00 | 1.00 | 1.00 |
| | 15 min | 0.95 | 1.00 | 1.00 | 1.00 | 1.00 |
| −20 | 5 min | 0.86 | 1.00 | 1.00 | 1.00 | 1.00 |
| | 10 min | 0.89 | 1.00 | 1.00 | 1.00 | 1.00 |
| | 15 min | 0.90 | 1.00 | 1.00 | 1.00 | 1.00 |

The three remaining cessation algorithms (Table 5; boldface) met all previous criteria during the process of elimination to be safe, reliable, time efficient, and have reasonable minimum and maximum advisory cancellation times. The three best-performing algorithms we tested were: (1) wait 15 min after $Z_H$ values drop below 40 dBZ at $-5\,^\circ$C, (2) wait 15 min after $Z_H$ values drop below 35 dBZ at $-10\,^\circ$C, and (3) wait 15 min after graupel dissipation at $-15\,^\circ$C. All three algorithms had a 100% success rate with no false alarms or misses. This means they would have kept the public safe by ending the advisory after lightning cessation for all 23 storms in this study. These three cessation algorithms also would have ended the lightning advisory between 10 and 45 min after cessation in all 23 cases with an average wait time of 24 min after the last flash. Thus, these algorithms improve upon the 30–30 rule by saving an average of 6 min per storm. Table 5 shows lightning advisory end times with the following ranges: (1) 11–45 min, (2) 10–44 min, and (3) 10–38 min after the last flash. This makes (3) graupel dissipation at $-15\,^\circ$C slightly better for time savings, although the three algorithms are comparable. In fact, all three algorithms ended lightning

advisories earlier than the 30–30 rule for 17 of the 23 storms, making them ~74% more effective at saving time.

**Table 3.** False Alarm Ratio is defined by the total number of false alarms divided by the total number of hits and false alarms: FAR = total false alarms/(total hits + total false alarms).

| Temp Level (°C) | Wait Time | dBZ ≥ 40 | dBZ ≥ 35 | Graupel | Graupel + dBZ ≥ 40 | Graupel + dBZ ≥ 35 |
|---|---|---|---|---|---|---|
| 0 | 5 min | 0.04 | 0.00 | 0.00 | 0.00 | 0.00 |
| | 10 min | 0.00 | 0.00 | 0.00 | 0.00 | 0.00 |
| | 15 min | 0.00 | 0.00 | 0.00 | 0.00 | 0.00 |
| −5 | 5 min | 0.00 | 0.00 | 0.00 | 0.00 | 0.00 |
| | 10 min | 0.00 | 0.00 | 0.00 | 0.00 | 0.00 |
| | 15 min | 0.00 | 0.00 | 0.00 | 0.00 | 0.00 |
| −10 | 5 min | 0.13 | 0.04 | 0.04 | 0.04 | 0.04 |
| | 10 min | 0.13 | 0.00 | 0.00 | 0.00 | 0.00 |
| | 15 min | 0.04 | 0.00 | 0.00 | 0.00 | 0.00 |
| −15 | 5 min | 0.23 | 0.04 | 0.04 | 0.04 | 0.04 |
| | 10 min | 0.09 | 0.04 | 0.00 | 0.00 | 0.00 |
| | 15 min | 0.05 | 0.00 | 0.00 | 0.00 | 0.00 |
| −20 | 5 min | 0.43 | 0.13 | 0.09 | 0.09 | 0.09 |
| | 10 min | 0.19 | 0.04 | 0.09 | 0.04 | 0.04 |
| | 15 min | 0.10 | 0.00 | 0.04 | 0.00 | 0.00 |

**Table 4.** The Critical Success Index is defined by the total number of hits divided by the total number of hits, false alarms, and misses: CSI = total hits/(total hits + total false alarms + total misses). Values in boldface represent a CSI of 1.00, which indicates algorithms with a 100% success rate of hits. The remaining values in gray represent algorithms with a CSI < 1.00. This means they had at least one false alarm and/or miss for the 23 storms examined.

| Temp Level (°C) | Wait Time | dBZ ≥ 40 | dBZ ≥ 35 | Graupel | Graupel + dBZ ≥ 40 | Graupel + dBZ ≥ 35 |
|---|---|---|---|---|---|---|
| 0 | 5 min | 0.96 | **1.00** | **1.00** | **1.00** | **1.00** |
| | 10 min | **1.00** | **1.00** | **1.00** | **1.00** | **1.00** |
| | 15 min | **1.00** | **1.00** | **1.00** | **1.00** | **1.00** |
| −5 | 5 min | **1.00** | **1.00** | **1.00** | **1.00** | **1.00** |
| | 10 min | **1.00** | **1.00** | **1.00** | **1.00** | **1.00** |
| | 15 min | **1.00** | **1.00** | **1.00** | **1.00** | **1.00** |
| −10 | 5 min | 0.87 | 0.96 | 0.96 | 0.96 | 0.96 |
| | 10 min | 0.87 | **1.00** | **1.00** | **1.00** | **1.00** |
| | 15 min | 0.96 | **1.00** | **1.00** | **1.00** | **1.00** |
| −15 | 5 min | 0.74 | 0.96 | 0.96 | 0.96 | 0.96 |
| | 10 min | 0.87 | 0.96 | **1.00** | **1.00** | **1.00** |
| | 15 min | 0.91 | **1.00** | **1.00** | **1.00** | **1.00** |
| −20 | 5 min | 0.52 | 0.87 | 0.91 | 0.91 | 0.91 |
| | 10 min | 0.74 | 0.96 | 0.91 | 0.96 | 0.96 |
| | 15 min | 0.83 | **1.00** | 0.96 | **1.00** | **1.00** |

**Table 5.** This table shows the range and average lightning advisory cancellation time (min) for each cessation algorithm. The format is as follows: minimum-maximum (average). Advisory cancellation times are based on the time of the last flash (t = 0). The values in boldface represent cessation algorithms with the best range of cancellation times (10–45 min after the lash flash). Values in gray represent algorithms with cancellation times that fell outside of this range. Cancellation times were not calculated for cessation algorithms with a CSI < 1.00 (missing values).

| Temp Level (°C) | Wait Time | dBZ $\geq$ 40 | dBZ $\geq$ 35 | Graupel | Graupel + dBZ $\geq$ 40 | Graupel + dBZ $\geq$ 35 |
|---|---|---|---|---|---|---|
| 0 | 5 min | | 4–46 (21) | 4–51 (25) | 4–51 (25) | 4–51 (25) |
| | 10 min | 5–47 (21) | 9–82 (29) | 9–56 (30) | 9–56 (30) | 9–82 (33) |
| | 15 min | 10–52 (26) | 14–87 (34) | 14–61 (35) | 14–61 (35) | 14–87 (38) |
| −5 | 5 min | 1–35 (14) | 4–44 (18) | 4–39 (19) | 4–39 (19) | 4–44 (20) |
| | 10 min | 6–40 (19) | 9–82 (25) | 9–44 (24) | 9–44 (24) | 9–82 (27) |
| | 15 min | **11–45 (24)** | 14–87 (30) | 14–49 (29) | 14–49 (29) | 14–87 (32) |
| −10 | 5 min | | | | | |
| | 10 min | | 5–39 (19) | 5–44 (23) | 5–44 (23) | 5–44 (23) |
| | 15 min | | **10–44 (24)** | 10–49 (28) | 10–49 (30) | 10–49 (28) |
| −15 | 5 min | | | | | |
| | 10 min | | | 5–33 (19) | 5–33 (19) | 5–41 (20) |
| | 15 min | | 5–46 (23) | **10–38 (24)** | 10–83 (27) | 10–46 (25) |
| −20 | 5 min | | | | | |
| | 10 min | | | | | |
| | 15 min | | 4–83 (20) | | 5–37 (20) | 5–83 (23) |

The least effective algorithms proved to be both unsafe and time inefficient (removed or gray; Table 5). In most cases, these algorithms had at least one false alarm where the advisory was ended prior to lightning cessation. The 5 min wait time proved to be the least effective, especially at −10 °C, −15 °C, and −20 °C for all radar thresholds. Results from Table 4 show that 16 out of 28 (57%) algorithms with CSI < 1.0 utilized a 5 min wait time. This short wait time provides little confidence to forecasters since the storm could just be weakening temporarily as lightning activity continues. The 40-dBZ reflectivity threshold also is one of the least effective algorithms due to false alarms and misses. That is, the 40-dBZ cessation algorithms are too lenient and accounted for 10 out of 28 (36%) algorithms with CSI < 1.0. This is more than any other radar parameter. There are two reasons for this: (1) A non-severe storm may never reach $Z_H \geq 40$ dBZ at a given temperature level, thereby producing a missed event or (2) Since the 40-dBZ echo descends in a storm before the 35-dBZ echo, there is a greater likelihood that this threshold will prematurely end an advisory, thereby producing a false alarm. Many other algorithms proved effective at ending lightning advisories with no false alarms or misses. However, they are not within the 10–45 min range or have an average wait time longer than 30 min (Table 5).

Overall, the performance metrics for cessation algorithms in this study were comparable to those tested in Florida [5]. This reinforces the utility of a wait time approach for lightning cessation guidance across different geographic regions. However, our results show that our best-performing cessation algorithms in Washington, D.C., end lightning advisories later than those in Florida [5]. A similar trend was found in [18] in which their lightning cessation model had a longer median lag-time for the Washington, D.C. area than found with the [19] model in Florida. The delay in ending lightning advisories for Washington, D.C., may be due to this area having stronger storms. That is, sea breeze convection in Florida tends to be on a smaller scale compared to large-scale synoptic boundaries that often propagate toward the Washington, D.C. area [22,23]. Stronger updrafts mean that

radar-based thresholds such as reflectivity and graupel presence are delayed in being met. This ultimately prevents the lightning advisory from ending sooner.

To summarize, this study examined lightning cessation algorithms for non-severe, isolated cells in the Washington, D.C. area. We tested combinations of three radar parameters ($Z_H$, graupel, graupel and $Z_H$) at five different temperature levels (0 °C, −5 °C, −10 °C, −15 °C, and −20 °C) using three different wait times (5, 10, and 15 min). This produced 75 cessation algorithms to test on 23 storms. Results suggest the cessation algorithms that best balance safety and time savings are: (1) wait 15 min after $Z_H$ values drop below 40 dBZ at −5 °C, (2) wait 15 min after $Z_H$ values drop below 35 dBZ at −10 °C, and (3) wait 15 min after graupel dissipation at −15 °C. These three cessation algorithms were the safest of those tested based on POD, FAR, and CSI skill scores. That is, they safely ended the advisory for all 23 storms after the last flash. These three algorithms also were time efficient, ending advisories for 17 of the 23 storms earlier than the 30–30 rule. Their average advisory cancellation time of 24 min after the last flash saved 6 min per storm compared to the 30–30 rule. However, further study is needed before recommending these three cessation algorithms for operational utility. They performed well on the 23 storms in this study, but a much larger sample size is required to increase confidence in their performance on future storms. Furthermore, our best-performing cessation algorithms are probably less effective for other types of convective cells (i.e., multicell, quasi-linear storms, severe storms) and during the cool season where isotherm heights crucial to storm electrification shift closer to the surface. This study showed that thresholds and wait times of cessation algorithms that worked well in Florida performed differently for storms in the Washington, D.C. area. Simulated lightning advisories also had delayed cancellations in Washington, D.C. Thus, cessation guidance from this study will likely need to be adjusted for other geographic regions. Examining a composite index was beyond the scope of this study, but previous cessation studies [18,19] using a bootstrap model show promising results. Therefore, we encourage future cessation studies to develop guidance using multiple radar thresholds and isotherm levels.

**Author Contributions:** Analysis and writing: J.J.D.; project development and administration: A.D.P.; review and editing: A.D.P. All authors have read and agreed to the published version of the manuscript.

**Funding:** There was no external funding source. We would like to thank Northern Vermont University-Lyndon for providing internal startup funds for this research.

**Institutional Review Board Statement:** Not applicable.

**Informed Consent Statement:** Not applicable.

**Data Availability Statement:** Lightning data from the Washington, D.C., Lightning Mapping Array can be obtained at https://lightning.nsstc.nasa.gov/lma/dclma/datacal.pl?class=pp&y=2018 accessed on 15 November 2021. NOAA Next Generation Radar Level II radar data can be downloaded from https://www.ncdc.noaa.gov/nexradinv accessed on 15 November 2021. The Warning Decision Support System–Integrated Information software can be downloaded after submitting a request at http://wdssii.org/download.shtml accessed on 15 November 2021. Storm data with detailed information for each convective cell (e.g., time, latitude, longitude) are available upon request.

**Conflicts of Interest:** The first author declares no conflict of interest. The second author serves as co-editor for this journal.

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
