# Peer review of "Lightning Cessation Guidance Using Polarimetric Radar Data and Lightning Mapping Array in the Washington, D.C. Area"

_atmosphere, doi:10.3390/atmos13071111_

Round 1
Reviewer 1 Report
I should point out that I did the review in the initial submission. After a first look at the article, I notice that the presentation of the sections is not satisfactory. There are extensive paragraphs that tire the reader. The writing style is not scientific, and the template of the article has many issues. Finally, the article does not have the proper structure (introduction, related work, materials and methods, results and discussion, conclusion) and flow.
Τhe authors deal with a subject area that has many aspects and fields of research. Polarimetric radar data and total lightning data are used to develop lightning cessation guidance for isolated cells in the Washington, D.C., area. The Hydrometeor Classification Algorithm is used to locate graupel for each convective cell.
In this submission, Results show that the three best-performing cessation algorithms are based on the status of (1) 40 dBZ at -5 Celsius, (2) 35 dBZ at -10 Celsius, and (3) graupel at -15 Celsius.
It lacks depth and novelty. It is not scientifically thorough in description and does properly cite literature where needed. The number of references is incomplete and not up-to-date.
The structure of the article should be described at the end of the introduction.
What are the main challenges in this domain? What are the limitations of the previous works which motivate the current study? Please evaluate how your study is different from others? Highlight the motivation of this research and summarize the challenges of previous studies.
The authors fail to provide a comparative analysis with previous studies based on features, datasets, or algorithms.
The discussion of the results is lacking, comparing the literature that supports each of the hypotheses with the results obtained in the empirical study;
The authors should summarize the limitations and the potential issues of this study. Also, the overall merit of the proposed approach should be highlighted.
Tables 2-5 could be included as supplementary material.
Reviewer 2 Report
The revised manuscript has been greatly improved. The points on methodology and results were better discussed. Based in that, I am recommending the publication of this manuscript in this format.
Round 2
Reviewer 1 Report
I have no additional remarks.
This manuscript is a resubmission of an earlier submission. The following is a list of the peer review reports and author responses from that submission.
Round 1
Reviewer 1 Report
In this article, the authors deal with a subject area that has many aspects and fields of research. Polarimetric radar data and total lightning data are used to develop lightning cessation guidance for isolated cells in the Washington, D.C., area. The Hydrometeor Classification Algorithm is used to locate graupel for each convective cell.
The article is well-written but, it has not the proper length. This submission is more appropriate for a conference than a scientific journal.
It lacks depth and novelty. It is not scientifically thorough in description and does properly cite literature where needed. The number of references is incomplete and not up-to-date.
The introduction section needs enrichment. In its current state, it lacks references to related studies. Also, the authors should mention, briefly, the main contribution of the work.
There is no specific methodology in this article. It has a simple application without any novelty. How your study is different from others?
The authors fail to provide a comparative analysis with previous studies based on features, datasets, or algorithms.
The results remain without compelling evidence of research novelty.
The discussion of the results is lacking, comparing the literature that supports each of the hypotheses with the results obtained in the empirical study;
The authors should summarize the limitations and the potential issues of this study. Also, the overall merit of the proposed approach should be highlighted.
Finally, what future directions emerge from this submission?
Reviewer 2 Report
Reviewer Evaluations:
The submitted article use dual-polarization radar and 3D LMA flash data for lightning cessation guidance for isolated storms over Washington, DC. This study it is very interesting, because has a practical application for lightning cessation nowcasting. I have some issues for this study: i) The results of manuscript needs be better physically discussed. ii) The quality of radar data and reprocessing methodology was not discussed in the text. iii) In addition, the results on skill scores showed in Table 2, needs be showed with numeric data. Based in that, I recommend MINOR corrections.
- Lines 20-21: “50 casualties and hundreds of injuries result from lightning strikes”. Please, be specific on which region you are discussing.
- Lines 22-23: “…cost effective for outdoor businesses”. This is the major importance of this study. Please, cite some study on theses impacts and losses.
- Lines 86-87: “...average temperature of 74oF while the Cape Canaveral area of Florida averages 80.3oF.”. This it an important issue on the meteorological differences between region. Please, insert an better and deep discussion on this point.
- Line 11. “b. Radar and Environmental Data”. Radar provides several products. Each product has an specific application. Did you have used Plan Position Indicator (PPI) or Constant Plan Position Indicator (CAPPI)?
- Lines 13-14. “…using similar criteria as [14]”. The methodology used to identify and classify the storm can impacts your results. In this way, explain in more details which methodology was used.
- Figure 2. This figure has poor resolution. Is not possible see the text in the image and the details.
- Figure 3. As you have analysed radar-lightning data spatially combined, insert in the same figure the radar distance range and the 150 km radius LMA network. Also insert the location of storms within this map.
- Lines 138-139. “We also ran the HCA algorithm (w2dualpol) [10] to determine the presence of graupel (Figure 4) as done in [14]”. The HCA algorithm used for hydrometeors classification it is very important issue, which can impact your results. Please, insert an explanation on this point.
- Figure 4. A better description of the figure caption it is necessary. Please, explain what are the yellow dots. Also define the names of hydrometeors acronyms. Again, explain the radar product utilized (CAPPI, PPI or other?).
- “The lightning products were then added to the WDSS-II software and overlaid on radar data (Figure 5)”. Which the temporal frequency of radar? The lightning flashes from LMA network was combined considered which time of accumulation?
- Figure 5. Explain in more details the legend of this figure, also improve the quality of figure.
- Line 159. The definition of “nonsevere” storms needs be explained. How long did these storms last?
- Line 160. “The advisory begins when the first lightning flash is detected”. Is this the first flash of storm lifecycle? The majority flash are which type, intracloud or cloud-to-ground flashes?
- Lines 203-205. “The existence of ZH ≥ 40 dBZ was also one of the least effective algorithms to use at these temperature levels due to false alarms and misses. This is because of a lack of greater reflectivity measured higher in the storm.”. This sentence is so confused. The manuscript needs be better physically discussed. Do a deep discussion why why this threshold was also one of the least effective.
Reviewer 3 Report
Lightning Cessation Guidance Using Polarimetric Radar Data 2 and Lightning Mapping Array in the Washington, D.C., Area
By Drugan and Preston
In the present manuscript, the authors reported the development of lightning cessation guidance for isolated cells over Washington, D.C., area. They utilized Polarimetric radar and three dimensional total lightning data, overlaid on a Warning Decision Support System - Integrated Information (WDSS-II) platform. Hydrometeor Classification Algorithm is also being used to locate graupel for each convective cell. For the present study, they considered 23 nonsevere thunderstorms during the 2015-2017 warm seasons.
They have reported that, the lightning cessation advisory ends after waiting a certain amount of time without meeting the following radar-based thresholds: 1. ZH ≥ 40 dBZ 2. ZH ≥ 35 dBZ 3. presence of graupel 4. presence of graupel with ZH ≥ 40 dBZ 5. presence of graupel with ZH ≥ 35 dBZ. These thresholds were tested at five different temperature levels (0 o C, -5 o C, -10o C, - 15o C, and -20o C) for three different wait times (5, 10, and 15 min) for a combination of 75 total algorithms. The results shows that the two best algorithms are (1) to wait 15 min after ZH values drop below 40 dBZ at the -5 o C level of a storm and (2) to wait 15 min after graupel dissipation at the -15 oC level of a storm. These two algorithms both had a 100% success rate (n = 23 storms) with no false alarms or misses. it is reported that, the top performing algorithm have improved to around 74% compared to 30-30 rules. These algorithms are recommended only for isolated cells in the region.
Major comments
The present work though is relevant and timely taken with the help of existing radars and three dimensional Lightning Mapping Array data on WDSS-II platform, yet it suffers from certain limitations.
(i). The major issue is lack of clarity in the result section, particularly with Table 2. The authors have just provided the results without a support of skill score. Author should provide the following skill score in a table form to support their statement for each algorithm.
POD- (Probability of detection/cessation); FAR -(False Alarm Ratio); CSI -(Critical Success Index);
HSS- (Heidke skill score) etc. The formulation details of these parameters should be provided in the methodology section.
(ii). I feel that, the present five radar criteria would be more suitable for prediction of lightning occurrence. In the present context of advisory of cessation the concluding statement is always indirect, where authors have to say essentially (Line 161) : “without meeting the following radar based criteria”. Contrary to this, for the prediction/guidance of cessation of lightning, the following criteria would be more direct and realistic and easy to comprehend 1. ZH < 40 dBZ 2. ZH < 35 dBZ 3. absence of graupel 4. absence of graupel with ZH < 40 dBZ 5. absence of graupel with ZH < 35 dBZ. In Table 2 providing the suggested criteria will be more direct and logical, otherwise the Table-2 is confusing.
(iii) Further, I feel, that authors should look into a composite index by taking into account, either all the thresholds or the most suitable identified thresholds, at five given temperature and for three different waiting time. In this way, no of algorithm may drastically reduce, with much improved results.
(iv). There is no discussion of their result in context of the previous work. The authors must discuss their results.
(v) After the discussion, authors should spell out the limitation of present approach and implication of the present work for future.
Minor comments
(i) Abstract (Line-13): “Results show that the two best-performing cessation algorithms utilize the presence of 40 dBZ 13 at -5 o C and graupel at -15o C.” The statement is a bit confusing. Instead, it should be like “Results show that the two best-performing cessation algorithms IS BASED ON THE STATUS of 40 dBZ at -5 o C and graupel at -15o C”
(ii). The statement (Line-160) : “The advisory begins when the first lightning flash is detected” . Plz clarify, what type of lightning flash is considered ? cloud to ground or intra-cloud.
In the present form, the manuscript is not suitable for publication. I recommend it for major revision.
.